# Beneficial and Detrimental Roles of Heme Oxygenase-1 in the Neurovascular System

**DOI:** 10.3390/ijms23137041

**Published:** 2022-06-24

**Authors:** Yoon Kyung Choi, Young-Myeong Kim

**Affiliations:** 1Bio/Molecular Informatics Center, Department of Bioscience and Biotechnology, Konkuk University, Seoul 05029, Korea; 2Department of Molecular and Cellular Biochemistry, School of Medicine, Kangwon National University, Chuncheon 24341, Korea

**Keywords:** heme oxygenase, carbon monoxide, iron, bilirubin, ferroptosis, regeneration

## Abstract

Heme oxygenase (HO) has both beneficial and detrimental effects via its metabolites, including carbon monoxide (CO), biliverdin or bilirubin, and ferrous iron. HO-1 is an inducible form of HO that is upregulated by oxidative stress, nitric oxide, CO, and hypoxia, whereas HO-2 is a constitutive form that regulates vascular tone and homeostasis. In brains injured by trauma, ischemia-reperfusion, or Alzheimer’s disease (AD), the long-term expression of HO-1 can be detected, which can lead to cytotoxic ferroptosis via iron accumulation. In contrast, the transient induction of HO-1 in the peri-injured region may have regenerative potential (e.g., angiogenesis, neurogenesis, and mitochondrial biogenesis) and neurovascular protective effects through the CO-mediated signaling pathway, the antioxidant properties of bilirubin, and the iron-mediated ferritin synthesis. In this review, we discuss the dual roles of HO-1 and its metabolites in various neurovascular diseases, including age-related macular degeneration, ischemia-reperfusion injury, traumatic brain injury, Gilbert’s syndrome, and AD.

## 1. Overview of Heme Oxygenase

Heme oxygenase, in the form of HO-1 and HO-2, is an essential enzyme in heme catabolism that cleaves heme to carbon monoxide (CO); biliverdin, which is rapidly converted to bilirubin; and ferrous iron (Fe^2+^). Oxygen (O_2_) is required as a co-substrate in this process [1]. HO-1 is strongly induced in various cells in response to hypoxia and stress and promotes neuroprotection and angiogenesis in the ischemic milieu [2]. In contrast, HO-2 is constitutively expressed in the neurons, where it functions as an intrinsic protector [3]. HO-2 is also the primary enzyme contributing to CO generation in the carotid bodies; hypoxia leads to a graded reduction in the CO levels in the carotid bodies [4]. CO is absent in HO-2-deficient carotid bodies [5].

The cells in HO-1-deficient mice demonstrate reduced stress defenses against hemin and hydrogen peroxide [6]. Furthermore, HO-1-deficient mice exhibit enhanced oxidative stress, tissue injury, and chronic inflammation, in addition to iron accumulation in the hepatic and renal cells [7]. HO-1-deficient mice also exhibit hypoxia-induced severe right ventricular dilation and infarction. Moreover, HO-1 deficiency exacerbates the formation and vascular remodeling of atherosclerotic lesions [8,9]. In humans, HO-1 deficiency presents as hemolysis, nephritis, and asplenia [10]. In a previous case report, oxidative stress caused increased endothelial cell injury in a 6-year-old boy with HO-1 deficiency [11], who presented with growth retardation and anemia [11].

Several HO-1 transcription factors exist, including nuclear factor erythroid 2-like 2 (Nrf2), biliverdin reductase (BVR), nuclear factor kappa-light-chain-enhancer of activated B cells (NF-κB), activator protein (AP)-1, and AP-2 [12,13]. The Nrf2–HO-1 system is an evolutionarily conserved mechanism involved in development, oxidative stress responses, and anti-inflammation [14]. Appropriate HO-1 levels exert beneficial effects, whereby Nrf2 functions as an adaptive response to oxidative stress [14]. However, abnormal HO-1 levels with Nrf2 dysfunction are implicated in the pathogenesis of neurovascular systems related to ischemia, trauma, and aging [12].

HO-1 downstream signals modulate angiogenesis, neurogenesis, and mitochondrial functions via vascular endothelial growth factor (VEGF) and its related factors, such as nitric oxide (NO) [15,16]. HO-1 also interacts with the NO-synthesizing enzyme NO synthase (NOS). The NOS family includes three distinct enzymes encoded by separate genes: neuronal NOS (nNOS (*NOS*-*I*)), inducible NOS (iNOS (*NOS*-*II)*), and endothelial NOS (eNOS (*NOS*-*III*)). Among the HO-1 metabolites, reciprocal synthesis between CO and NO occurs via crosstalk with the HO-1–CO and NOS–NO axes [17]. Regulation of gene expression and modification of protein thiols by NO further regulate the HO-1-mediated cellular functions [18].

In this review, we focus on the dual roles and underlying molecular mechanisms of HO-1 in the neurovascular unit. The HO system and its metabolites play various functions in neurovascular diseases, such as age-related macular degeneration (AMD), ischemia-reperfusion (IR) injury, traumatic brain injury (TBI), and Alzheimer’s disease (AD). In the core injury region of the brain, cellular damage can be exacerbated by excessive HO-1 metabolites in an oxidative stress milieu. In the peri-injured region, regenerative and repair mechanisms can be promoted by moderate levels of HO-1 metabolites in an antioxidant milieu (Figure 1).

## 2. HO Metabolites

HO metabolites include CO, biliverdin, and iron. Biliverdin can be converted into bilirubin by BVR. These HO metabolites have a “Janus face”. Despite their pathophysiological role as antioxidants, they are toxic at higher concentrations (Figure 2). In this section, the dual roles of the HO metabolites in the neurovascular system are discussed.

### 2.1. CO

HO-derived CO interacts with Fe^2+^ of heme-containing proteins, including soluble guanylate cyclase and cytochrome *c* oxidase, leading to various cellular events. CO, generated either by exogenous delivery or HO activity, is fundamentally involved in the regulation of mitochondria-mediated redox cascades for adaptive gene expression and increasing blood circulation (i.e., O_2_ delivery), via vasorelaxation and neovascularization, underscoring the regulation of mitochondrial energy metabolism and blood circulation.

The blood barriers between the blood and neural tissues are collectively referred to as the blood–neural barrier and comprise the blood–brain, blood–retinal, blood–labyrinth, and blood–cerebrospinal fluid barriers [19]. These barriers play an essential role in protecting the nervous system from harmful materials and agents. Blood barrier integrity is regulated by the HO–CO axis. In this regard, CO gas and pharmacologically CO-releasing molecules (CORMs) protect against ischemic stroke by reducing neuroinflammation through the alleviation of blood–neural barrier disruption [20,21,22,23]. In the cerebral microvascular endothelial cells (CMVECs) of newborn pigs, excitotoxic concentrations of glutamate released from the neurons cause oxidative stress-mediated apoptosis and disruption of the blood–neural barrier in an in vitro model of cerebral vascular endothelial injury via mitochondria-mediated superoxide generation [20]. This suggests that glutamate receptor (GluR) activation and mitochondrial reactive oxygen species (ROS) production are involved in the mechanisms of cerebral endothelial cell dysfunction. Notably, CORM-A1 exhibits potent antioxidant and antiapoptotic properties in CMVECs, and completely prevents blood–neural barrier dysfunction caused by glutamate and iGluR agonists [21]. However, the CO poisoning, induced by high concentrations or long durations of CO exposure, results in tissue hypoxia and damage, leading to blood–neural barrier disruption and interstitial fluid accumulation, with decreased circulating blood volume [24].

### 2.2. Biliverdin and Bilirubin

Biliverdin, the first product of heme cleavage by HO, triggers Ca^2+^/calmodulin-dependent protein kinase (CaMK) signaling, eNOS phosphorylation, and iNOS-independent NO production in macrophages [18]. The NO nitrosylates BVR, leading to the translocation of BVR into the nucleus, where it binds to the toll-like receptor (TLR)-4 at AP-1 sites to block transcription [18]. The BVR comprises two isoforms: BVR-A and BVR-B. BVR-A reduces the HO-derived biliverdin into bilirubin [25], but the specific functions of BVR-B in adults have not been elucidated.

Bilirubin has dual roles as an antioxidant and toxic molecule [26]. Similar to CO, bilirubin acts as a toxin at high concentrations (higher than approximately 300 μM), especially in neonates, leading to mitochondrial dysfunction and cell death [26]. Moderate concentrations of bilirubin have beneficial effects, such as in regeneration and anti-inflammation in neurovascular diseases. Bilirubin binds to NO (bilirubin-NO) via *N*-nitrosation reactions and possesses anti-nitrosative capability by scavenging NO [27]. The physiological levels and antioxidant effects of bilirubin are dependent on the expression and activity of BVR, which converts biliverdin into bilirubin [28]. However, high levels of bilirubin (300 μM) induce an uncoupled oxidative phosphorylation and cytochrome *c* release in rat-liver isolated mitochondria via perturbation of the mitochondrial membrane function, leading to intracellular energy failure [29]. Thus, abnormal plasma or serum bilirubin levels may serve as a biomarker for the risk of developing Gilbert’s syndrome [30].

### 2.3. Iron

Iron plays a key role in the electron transfer between Fe^2+^ and ferric iron (Fe^3+^) through oxidation–reduction (redox) reactions. Labile iron produces ROS, including hydroxyl radicals (OH), by a reaction between Fe^2+^ and hydrogen peroxide, termed the Fenton reaction [31,32]. Iron is mostly bound to ferritin (a sequestering protein) or enters into neurons and astrocytes in the brain through transporters, such as divalent metal transporter 1 (DMT1) [33]. Non-transferrin-bound iron (NTBI) has been identified in the plasma of patients with chronic iron overload disorders, such as hemochromatosis and thalassemia, in which transferrin saturation is significantly elevated [34]. The plasma NTBI is composed predominantly of Fe^3+^ and is loosely bound to the buffering molecules (mainly citrate and acetate), as well as the abundant plasma protein albumin [33,35]. Excessive intracellular labile iron leads to ferroptosis, a form of iron-dependent lipid peroxidation, and consequent cell death in the brain cells, which underscores the pathology of several neurodegenerative diseases [32]. A critical morphological feature of ferroptosis is mitochondrial dysfunction, characterized by smaller mitochondrial volume and the rupture of the mitochondrial outer membrane in ferroptosis-processing cells compared to that in normal cells [36].

The enhanced HO-1 expression and activity increase the levels of free iron and subsequent expression of ferritin [37,38,39]. Alongside ferritin production, ferroptosis induction by HO-1 may be associated with the reduced free iron-binding ability of ferritin induced by oxidative stress, such as iron accumulation and lipid peroxidation [40,41]. The sustained overexpression of HO-1 in astrocytes leads to abnormal iron deposition and mitochondrial dysfunction in the brain, resulting in decreased cognitive ability [37,42]. HO-1 expression results in the rapid expression of ferritin [39] and an ATPase pump [43] that actively removes intracellular iron, thereby exerting cytoprotective effects. HO-1 regulates ferroptosis according to the cell type (e.g., cancer or normal cells) and context (e.g., a Nrf2-dependent or Nrf2-independent milieu) [40,44,45]. In normal cells, excessive iron production due to chronic HO-1 overexpression likely facilitates intracellular toxicity and cell death (ferroptosis); conversely, appropriate levels of HO-1, induced by Nrf2, protect the cells from iron-dependent toxicity [12,37]. 

Biological and cellular iron homeostasis is regulated by various proteins, including iron regulatory proteins (e.g., IRP1 and IRP2), ferritin, transferrin, ferroportin1, and DMT1 [46]. Transferrin, a monomeric serum glycoprotein, binds to two Fe^3+^ ions and delivers iron into the cellular endosomes through receptor-mediated endocytosis. The acidic milieu drives the dissociation of iron from the transferrin receptor complex. After the release of iron, apo-transferrin is recycled to the plasma membrane, where it is released from its receptor to scavenge more iron [47,48]. IRP1 and IRP2 are cytosolic RNA-binding proteins that sense the iron concentration and post-transcriptionally regulate the expression and translation of the iron-related genes, to maintain cellular iron homeostasis [49]. Ferroportin 1 is the only identified mammalian nonheme iron exporter. Ferroportin 1 exports intracellular iron into the bloodstream to maintain appropriate iron homeostasis [50]. Excessive iron stimulates ROS and reactive nitrogen species (RNS) production and lipid peroxidation, whereas appropriate iron recycling exerts protective effects on the neurovascular system. Indeed, iron homeostasis plays key roles in mitochondrial function, redox states, and inflammation in the various diseases discussed in Section 4.

## 3. Signaling Cascades Related to HO-1

Biological molecules related to HO-1 expression and function, including upstream (e.g., Nrf2 and BVR) and downstream signals of HO-1 (e.g., peroxisome proliferator-activated receptor-γ [PPARγ] coactivator-1 (PGC-1), estrogen-related receptor (ERR), and hypoxia-inducible factor-1 (HIF-1)), as well as crosstalk between HO-1–CO and NOS–NO are discussed in this section. HO-1 and its related signaling pathways contribute to various antioxidant, anti-inflammatory, and regenerative effects.

### 3.1. Upstream HO-1 Signals

CO stimulates Nrf2 and HO-1 expression [51,52], resulting in a positive feedback loop of Nrf2–HO-1–CO. HO-1 regulates regeneration, anti-inflammation, and iron sequestration via the Nrf2 pathway. Interaction between the Nrf2–HO-1–PGC-1α signaling pathways improves the mitochondrial function and biogenesis [53,54]. The treatment of microglia with an alpha-7 nicotinic acetylcholine receptor (nAChR) agonist (PNU282987) induces mitochondrial biogenesis, which is abolished in the microglia obtained from Nrf2^−/−^ HO-1-mutant mice, or PGC-1α^−/−^ mice [55]. Reciprocal activating crosstalk among the HO-1, CO, and PGC-1α pathways also induces the expression of various genes involved in mitochondria biogenesis, angiogenesis, neuroprotection, and neurogenesis, by activating transcription factors such as ERRα and HIF-1α [54,56,57,58].

BVR-A translocation from the cytosol to the nucleus is essential for regulating HO-1 gene expression by hematin [13]. The nucleus BVR-A acts as a leucine zipper-like DNA-binding protein and functions as a transcription factor for AP-1 and cyclic AMP-regulated genes, resulting in the transcriptional induction of activating transcription factor-2 and HO-1 [59,60]. In sum, the BVR-A–HO-1 axis facilitates the endogenous bilirubin production and plays a major role in cytoprotective effects.

### 3.2. PGC-1α as a Downstream Signal of HO-1 

Mitochondria serve as the cellular powerhouse that generates ATP or heat-using substrates derived from fat and sugar. Mutations in the mitochondrial or nuclear DNA that affect the mitochondrial oxidative metabolism underscore endocrinal, ophthalmological, and neurodegenerative disorders [61,62]. Given the protective role of HO-1 against the mitochondrial dysfunction associated with aging and the development of various clinical symptoms [63], we emphasize the importance of HO-1-mediated cellular processes and signaling pathways linked to PGC-1α.

PGC-1 proteins function as the master regulators of mitochondria biogenesis and energy production [64], and play key roles in VEGF expression and angiogenesis [54,57]. PGC-1α has a complex multidomain structure that is associated with its key role in the regulation of various cellular processes via the co-activation of nuclear receptors and transcription factors, including ERRα and FOXO1. PGC-1α interacts with >20 transcription factors, including ERRs [57]. PGC-1β has a high sequence and expression pattern similarity to PGC-1α, with a high expression in the high-metabolic rate organs, such as brown adipose tissue, the heart, and slow-twitch soleus muscle. PGC-1β-deficient mice exhibit altered expression of many of the mitochondrial energy metabolism genes, are less sensitive to acute cold exposure, and develop hepatic steatosis when fed a high-fat diet [65]. In skeletal muscle, PGC-1α activation induces increased mitochondrial biogenesis through AMP-activated protein kinase (AMPK)-mediated phosphorylation and AMPK-NAD^+^-sirtuin 1 (SIRT1)-mediated deacetylation [66,67]. In addition, the activation of AMPK increases VEGF-mediated angiogenesis by activating PGC-1α in a mouse model of hindlimb ischemia [66,68].

The ERR subfamily, comprising ERRα, ERRβ, and ERRγ, constitute a key partner for mitochondrial oxidative metabolism [57]. ERRγ overexpression largely rescues autonomous muscle injury or damage in a muscle-specific PGC-1α/β double-knockout (KO) mouse model [69]. Further, ERRγ binds to the VEGF promoter region and increases *VEGF* gene expression, independent of PGC-1α/β expression in the plantaris [69]. However, the transcriptional activity of ERRα is largely dependent on the presence of PGC-1α, which is expressed at low basal levels under normal conditions and is induced by energy stress [70].

Ca^2+^ is a major contributing factor to intracellular signal transmission, based on changes in Ca^2+^ with energy demands. Of note, Ca^2+^-mediated signaling by CO is cell type-dependent. CO inhibits cardiac L-type Ca^2+^ channel function through the redox modification of three cysteine residues in its cytoplasmic C-tail region by increasing the formation of ROS from mitochondrial respiratory complex III [71]. In contrast, CO promotes the Ca^2+^ influx via N-type and L-type Ca^2+^ channels in the carotid body [72,73]. Moreover, CO activates the intestinal L-type Ca^2+^ channels by activating eNOS, but not iNOS or nNOS [74]. The Ca^2+^ chelators inhibit the CO-mediated mitochondrial biogenesis in astrocytes by suppressing PGC-1α/ERRα activation [75]. Both the PGC-1α expression and mitochondrial biogenesis in brain tissue are higher in wild-type mice than in HO-1^+/−^ mice [75]. These data suggest that the HO-1–CO axis promotes mitochondrial biogenesis via Ca^2+^-dependent PGC-1α/ERRα activation. In astrocytes exposed to CORM-2 (a CO-releasing molecule) or transfected with HO-1, VEGF expression and mitochondrial biogenesis are increased via the sequential activation of the following signal cascades: L-type voltage-gated Ca^2+^ channel-CaMKβ-mediated AMPKα activation; AMPKα-induced increases in nicotinamide phosphoribosyl transferase (NAMPT) expression and cellular NAD^+^/NADH ratio; SIRT1-dependent PGC-1α stabilization and activation; and PGC-1α/ERRα-mediated mitochondrial activation and VEGF expression [54,75]. The combination of CO and bilirubin further activates the Ca^2+^–NAMPT–NAD^+^/NADH–PGC-1α pathway [54]. Moreover, iron chelation attenuates the mitochondrial protein and transcript levels induced by PGC-1α in C2C12 mouse myoblasts [76], suggesting that PGC-1α-mediated mitochondrial biogenesis is regulated by the accumulation of cellular iron, a major HO-1 metabolite. Notably, the activation of nAChRs, a family of ion-gated channels, increases the mitochondrial biogenesis and function in microglial cells via the Nrf2–HO-1–PGC-1α axis [55]. Collectively, these results suggest that HO-1 improves mitochondrial biogenesis and vascular angiogenesis in a PGC-1α-dependent manner. 

### 3.3. HIF-α as a Downstream Signal of HO-1

HIFs transactivate > 1500 target genes by binding to the cis-acting hypoxia response elements containing the sequence 5′-(A/G)CGTG-3′, which are located within the target genes or their flanking sequences. In all of the cells, hypoxia increases the expression of hundreds of mRNAs and decreases the expression of a similar number of genes in a HIF-dependent manner [77]. *VEGF* is one of the genes upregulated by HIFs in various cell types and regulates angiogenesis and vascular density. Indeed, the HIFs function as master regulators of intracellular oxygen homeostasis by controlling vascular remodeling. HIFs are heterodimeric proteins that consist of an O_2_-regulated HIF-α subunit (HIF-1α, HIF-2α, or HIF-3α) and a constitutively expressed HIF-β subunit. HIFs undergo transcriptional, post-transcriptional, translational, and post-translational regulation. At normal O_2_ levels, iron-dependent proline hydroxylases (e.g., PHD1, PHD2, and PHD3) hydroxylate one or both of the proline residues (Pro402 and Pro564 on HIF-1α and Pro405 and Pro531 on HIF-2α) located in the O_2_-dependent degradation domain of the HIF-1α subunit, which is subsequently degraded via the proteasome pathway [78]. The PHD substrates include O_2_, iron, and 2-oxoglutarate; in this regard, iron chelation strongly induces HIF-α protein stability by inactivating iron-dependent PHDs and consequently diminishes HIF-α degradation [79]. The 5′-untranslated region (UTR) of HIF-2α-encoding mRNA contains an iron regulatory element (IRE) sequence similar to that of ferritin [80]. Sufficient intracellular iron leads to the assembly of iron-sulfur clusters (Fe-S) in the IRE-binding pocket of IRP1, which blocks its interaction with the IREs present in the 5′-UTR of ferritin and HIF-2α mRNAs. This results in an increase in the translational efficiency of both of the gene transcripts [79,81]. Therefore, high levels of intracellular iron stimulate HIF-2α expression.

In the absence of exogenous stimuli, HO-1–CO increases HIF-1α-dependent VEGF expression in astrocytes via dual mechanisms. First, CO promotes de novo protein synthesis of HIF-1α through the activation of the phosphoinositide 3-kinase (PI3K)–Akt–mammalian target of rapamycin and the MEK–ERK pathways responsible for the activation of the translational machinery. Second, CO stabilizes the HIF-1α protein by inhibiting proteasomal degradation through functional activation of heat shock protein 90 (Hsp90) and subsequent interaction with HIF-1α [82]. Hsp90α interacts directly with and protects HIF-1α from O_2_^−^ and PHD-independent ubiquitination and degradation [83,84], resulting in elevated HIF-1α protein levels and VEGF expression.

Astrocytes pre-exposed to CORM-2 induce PGC-1α and ERRα expression, resulting in increased mitochondrial function [75], and O_2_ consumption [56]. The stabilized HIF-1α binds to the ERRα promoter and increases its mRNA and protein levels [56]. In an in vivo ischemic brain injury model, enhanced HIF-1α protein immunoreactivity is observed in the peri-infarct region in wild-type mice, which is abolished in HO-1^+/−^ heterozygote mice. Experiments with various pharmacological inhibitors and siRNAs have revealed sequential signaling events, including HO-1-derived CO, HIF-1α protein, ERRα expression via L-type Ca^2+^ channels, Ca^2+^/CaMKβ-mediated activation of AMPKα, AMPKα-dependent HO-1 induction, and the consequent stabilization of HIF-1α in a PHD2-dependent manner [56]. Collectively, these findings indicate that HO-1-derived CO elicits a HIF-1α–ERRα circuit by activating L-type Ca^2+^ channels and Ca^2+^ influx.

Hypoxia (1% O_2_) induces HO-1 mRNA in rat aortic vascular smooth muscle in a HIF-1α-dependent manner [85]. The hypoxia-mediated increase in HIF-1α/DNA-binding activity is inhibited by CO in human hepatocellular carcinoma cells [86]. Although HIF-α is considered a downstream signaling mediator of HO-1, the relationship between HIF-α and HO-1 metabolites is dynamic, complex, and operates in an oxygen-dependent manner in various cell types. Notably, the mechanisms underlying these relationships warrant further investigation.

### 3.4. Interplay between HO-1 and NOS

The interplay between HO-1 and NOS modulates the inflammatory responses through the HO-1 metabolite-mediated regulation of NOS expression and activation. The HO metabolites, such as iron and CO, reduce NO production in inflammatory immune cells. Increasing intracellular concentrations of free iron decrease iNOS gene expression and NO formation in inflammatory murine macrophage cells. Decreased iNOS/NO activity results in a reduction of the IRE-binding activity of IRPs via the decreased formation of the iron-sulfur-nitrosyl complexes, thereby leading to increased ferritin translation and iron storage [87]. Further, CORM-2-derived CO inhibits iNOS expression in inflammatory murine macrophages via PPARγ activation [88]. CORM-2 inhibits NO production in BV2 microglia cells during lipopolysaccharide stimulation [89]. In contrast, CORM-2 restores the tumor necrosis factor (TNF)-α-induced eNOS/NO downregulation by inhibiting the NF-κB-responsive miR-155-5p expression in human umbilical vein endothelial cells. In sum, the HO-1–CO axis differentially regulates the iNOS and eNOS expression in macrophages and endothelial cells, respectively. In addition, CO reduces the production of ROS and RNS, consequently decreasing cytotoxic peroxynitrite generation and ischemic injury [90].

The activation of AMPK, a crucial cellular energy sensor, stimulates eNOS by phosphorylating it at Ser^1179^, highlighting the potential crosstalk between cellular metabolism and vascular tone. In experiments on flow channels in cultured endothelial cells, AMPKα phosphorylation was increased with changes in the flow rate or pulsatility [91]. Furthermore, the AMPK activity and eNOS phosphorylation are significantly elevated in the aortas of high-runner mice. The combination of CO and bilirubin enhances the AMPK phosphorylation and activity in astrocytes [54], implying an interaction between HO-1 and NOS. The treatment of pericytes with CORM-3 (200 μM) facilitates the repair and regeneration via activation of the nNOS–NO pathway in neural stem cells [23]. In PC12 cells, the bilirubin (0.5–10 μM) stimulates ERK1/2 and CREB phosphorylation and nNOS–NO production in the absence of exogenous growth factors; these effects are blocked by the extracellular Ca^2+^ chelator, ethylene glycol tetraacetic acid [92]. These data collectively indicate that the HO-1 metabolites evoke various pathophysiological events and cellular signaling cascades in an NO-dependent manner via the regulation of the iNOS/eNOS–NO pathway.

## 4. Dual Effects of HO-1 in Neurovascular Diseases

In this section, the signaling pathways related to the dual (beneficial vs. detrimental) functions of HO-1 in various neurovascular diseases are discussed. The transient or moderate levels of HO-1 upregulate antioxidants, such as CO and bilirubin. However, weakened antioxidant defense and mitochondrial dysfunction in pathophysiological conditions associated with neurovascular diseases enable the extensive and persistent expression of HO-1, and the consequent overproduction of cytotoxic free iron. An imbalance in the redox system in tissues and cells leads to oxidative stress, blood–neural barrier disruption, and neurotoxicity. The molecular mechanisms underlying these effects include oxidative stress, ferroptosis, and chronic inflammation, and may be associated with a reduction in the regenerative potential of the neurovascular unit (Figure 1).

### 4.1. AMD

AMD is the most common cause of vision loss in individuals aged >65 years. The underlying pathology involves the degeneration of photoreceptors after a gradual disappearance of the retinal pigment epithelium. The pigment epithelium plays a crucial role in the barrier function (outer blood–retinal barrier) of the retinal photoreceptors [93]. The capillaries in the choroid underlying the pigment epithelium are the primary source of nourishment for the retinal photoreceptors. In neovascular AMD, abnormal blood vessel growth occurs under the macula in a hypoxic milieu [94]. ROS-induced HIF-1α stabilization increases VEGF expression, leading to a permeable outer blood–retinal barrier [94,95,96]. The increased choroidal vessel permeability results in fluid and blood leakage into the retina, ultimately causing damage to photoreceptors. The choroid vasculature exhibits numerous fenestrations that allow the passage of large molecules into the extravascular space and is highly susceptible to oxidative stress. In this regard, aging-related oxidative stress and iron accumulation may contribute to the AMD pathogenesis through ferroptosis [97]. The iron and transferrin levels in the retinal pigment epithelium (RPE) of post-mortem retinas are higher in patients with AMD than in age-matched controls [98,99]. Moreover, the interdependence of Nrf2 and HO-1 for ferroptosis mitigation is implicated in AMD repair [100,101,102,103,104].

Aged Nrf2 KO mice exhibit human AMD-like degenerative features in the RPE, such as Drusen-like deposits, accumulation of lipofuscin, spontaneous choroidal neovascularization, and sub-RPE deposition of inflammatory proteins [105]. This suggests that Nrf2-dependent antioxidant target genes, including *HO*-*1* and phase II antioxidant enzymes (e.g., NQO-1, GST, GCL, catalase, and SOD2), may suppress AMD pathogenesis. The 25129A>C polymorphism of the *Nrf2* gene is strongly associated with the occurrence of age-related macular degeneration [106]. Notably, the 19G>C polymorphism in the *HO*-*1* gene is observed in patients with AMD, indicating that the genetic variation of HO-1 may be associated with AMD pathogenesis via the modulation of the cellular reaction to oxidative stress [107]. A recent study has reported that RPE degeneration, a major pathologic process of AMD, is associated with ferroptosis via HO-1 induction and ferrous ion accumulation in the sodium iodate-induced oxidative stress model, and this process is inhibited by siRNA-mediated HO-1 knockdown or administration of the HO-1 inhibitor ZnPP or the iron chelator, deferoxamine [108]. These findings imply that oxidative stress-induced HO-1 overexpression is detrimental to retinal degenerative diseases, including AMD, via intracellular iron accumulation. Although the Nrf2–HO-1 system protects photoreceptors by suppressing RPE cell damage and degeneration, the beneficial effects of HO-1 and its metabolites have not been investigated in detail. In addition, oxidative stress-induced HO-1 overexpression underscores AMD pathogenesis by inducing ferroptosis. Nevertheless, further studies are required to clarify the relationship between HO-1 and AMD pathogenesis.

### 4.2. IR Injury

IR injury may occur in the central nervous and cardiac systems after a stroke. Persistent HO-1 expression in the brain is detected in human focal cerebral infarctions [109]. Iron accumulation has been implicated in long-lasting HO-1 expression in stroke. Gill et al. reported a correlation between the iron status and risk of stroke, and demonstrated that increased cardioembolic stroke risk was associated with higher serum iron and lower serum transferrin (an indicator of lower iron status) levels [110]. Furthermore, prolonged ischemia induces neuronal death in the hippocampal cornu ammonis 1 (CA1) region [111]. During ischemia, the neurons are challenged by excess extracellular glutamate in the presence of high levels of extracellular iron. This event causes glutamate receptor overactivation that boosts neuronal iron uptake and subsequent overproduction of lipid peroxides, leading to neuronal ferroptosis [47].

Beneficial effects of HO-1 and its metabolites in IR injuries have been reported. HO-1 is upregulated in the Müller cells of the rat retina and promotes cell survival after IR injury, which is abolished by the intravitreal injection of HO-1 siRNA before ischemia [112]. A HO-1 siRNA injection increases macrophage infiltration and severe destruction of the retinal structure in a rat model of IR injury [112]. Further, HO-1-mediated antioxidant properties, mediated by the generation of bilirubin and CO, may contribute to its beneficial effects on ischemic injury. Administration of soluble bilirubin nanoparticles using polyethylene glycol-modified bilirubin exerts potential therapeutic effects in a mouse model of IR injury [113]. Mice with depleted endogenous bilirubin via BVR deletion are susceptible to excitotoxic neuronal death, due to a decrease in antioxidant activity, particularly to superoxide radicals [114]. The therapeutic effects of CO have been demonstrated in retinas with IR injury. In rats, intravitreal injections of a CORM (ALF-186) protect the retinal ganglion cells from IR injury by suppressing the expression of inflammatory and apoptotic genes (i.e., NF-κB, TNF-α, and caspase-3) and promoting the expression of growth-associated protein 43, a marker of neural outgrowth and regeneration [115]. Indeed, the HO-1–CO pathway elicits protective and regenerative effects in the retinal Müller and ganglion cells after IR injury. 

In mice, inhalation of low-concentration CO gas (250 ppm) for 18 h immediately after ischemia stimulates Nrf2 translocation to the nucleus and upregulates HO-1 expression in brain tissue, compared to that in air-exposed animals, resulting in decreased infarct size and neurobehavioral function after a permanent focal cerebral ischemia. However, these effects are not observed in the Nrf2-deficient mice [51], indicating that the CO–Nrf2–HO-1 pathway plays a critical role in protection against cerebral IR injury. The HO-1–CO axis increases angiogenesis by upregulating VEGF expression in astrocytes [82], implicating the HO-1–CO axis in neuroprotection via angiogenesis in the ischemic brain. 

The HO-1–CO axis induced by IR injury increases angiogenesis via PGC-1α–ERRα-mediated and HIF-1α-dependent VEGF expression in astrocytes [56]. Angiogenesis, mitochondria biogenesis, and neurogenesis are stimulated by astrocyte-derived VEGF [75,82]. Additionally, the HO-1 pathway increases brain-derived neurotrophic factor (BDNF) expression, which is a key regulator of neuroprotection and regeneration [116]. BDNF and its cognate receptor, tropomyosin-related kinase B (TrkB), are expressed in the developing mouse cerebral cortex [117]. HO-1 overexpression in the hippocampal CA1 region 5 days before cerebral IR injury protects against hippocampal neuronal injury via activation of the BDNF–TrkB–PI3K signaling pathway [118]. Collectively, the regenerative and protective functions of HO-1 in IR injury appear to be associated with an increase in the expression of neurotrophic and anti-apoptotic genes and downregulation of inflammatory cytokines.

### 4.3. TBI

TBI is a primary disruption of the normal brain function, followed by secondary events including hemorrhage, edema, blood–brain barrier disruption, impaired mitochondrial function, and inflammation [119,120]. In humans, HO-1 immunoreactivity can be detected in microglia or macrophages until 6 months post-TBI [109]. In a TBI mouse model, iron deposition is increased until 28 days in the ipsilateral hemisphere, particularly in the core-injured region, compared to that in the contralateral hemisphere in both control young (3-month-old) and aged (24-month-old) mice [121,122]. Iron is upregulated by hemorrhage or microbleed-mediated heme degradation. In addition, the damaged endothelial cells, microglia, astrocytes, oligodendrocytes, and neurons facilitate iron accumulation post-TBI, due to excessive ROS and RNS production, imbalanced redox states, disruption of the iron-trapping protein expression, and enhanced mitochondrial dysfunction [37,123].

Experiments involving the administration of HO-1 metabolites or inducers, including CO, in TBI mice have revealed the beneficial effects of HO-1. Indeed, 1 h after TBI, the CORM-3-treated mice exhibit significantly lower MMP-9 levels in the damaged cortex and hippocampus compared to the vehicle-treated mice, and these effects are also observed in TBI mice treated with *N*-*tert*-butyl-α-phenylnitrone (a ROS and RNS scavenger) [23]. In addition, treatment with CORM-3 or CO gas inhalation significantly reduces the amount of pericyte cell death and promotes neurogenesis in a mouse model of TBI. Crucially, these protective effects are diminished by the administration of an HO inhibitor (tin protoporphyrin IX (SnPP)) [23]. Therefore, the reciprocal signaling circuits involving the HO-1–CO and CO–HO-1 axes play a crucial role in protection and neurogenesis after TBI [23]. Notably, administration of Korean Red Ginseng upregulates HO-1 expression in astrocytes of the peri-injured region 3 days after TBI, which is strongly correlated with the promotion of astrocytic mitochondrial function (e.g., ATP levels and O_2_ consumption). However, these effects are inhibited by SnPP treatment immediately before TBI [124]. The protective effects of HO-1 induction on TBI may involve the stimulation of mitochondrial biogenesis through the AMPKα–PGC–1α–ERRα circuit-mediated pathways [124] or translocase of outer mitochondrial membrane 20 pathway [125]. Further, the enhancement of mitochondrial quality and quantity control in astrocytes via the HO-1-dependent signaling pathway may modulate improvements in neural stem cell (NSC) function and neurogenesis in the peri-injured region after TBI [125].

The cellular network, between pericytes and endothelial cells, plays a major role in the blood–neural barrier formation, maintenance, and repair [19,126]. In this regard, CORM-3 reduces pericyte apoptosis and vascular leakage after TBI, possibly by decreasing HIF-1α expression and ROS and RNS production [23]. In addition to anti-permeability effects, CORM-3 treatment enhances nNOS activity in NSCs and neurogenesis in the peri-injured brain regions of mice after TBI, compared to that in the control (saline-treated) mice. Notably, these beneficial effects are reduced by treatment with a NOS or HO inhibitor [23]. Further, the conditioned medium obtained from pericytes treated with CORM-3 under oxygen-glucose deprivation conditions has the potential to stimulate in vitro differentiation of NSCs into mature neurons [23]. Additionally, plasma levels of bilirubin and antioxidant activity are elevated in patients with TBI [127]. Collectively, these findings suggest that CO and bilirubin exert beneficial effects on TBI by preventing acute pericyte apoptosis, rescuing pericyte crosstalk with NSCs, and promoting delayed neurogenesis after TBI.

### 4.4. Gilbert’s Syndrome

Hyperbilirubinemia is classified as having predominantly unconjugated bilirubin (indirect bilirubin) or conjugated bilirubin (direct bilirubin). Neonatal hyperbilirubinemia results from increased total serum bilirubin and clinically presents as pathologic conditions, such as jaundice, as well as behavioral and neurological impairments [128]. Gilbert’s syndrome is characterized by elevated unconjugated bilirubin levels in the blood due to increased bilirubin production or decreased bilirubin clearance [30,128]. The unconjugated bilirubin is transformed to conjugated bilirubin by the activity of hepatic uridine diphosphate-glucuronosyltransferase 1A1 (UGT1A1) to increase its water solubility [129]. Defective UGT1A1 can be a cause of Gilbert’s syndrome.

Bilirubin-induced neurotoxicity may occur when the central nervous system is chronically exposed to high levels of unconjugated bilirubin. Indeed, patients with Gilbert’s syndrome have higher concentrations of unconjugated bilirubin, carboxyhemoglobin, and iron than healthy controls [130]. Nonetheless, HO-1-mediated bilirubin production may have a beneficial role in various pathologic conditions [129]. Moderate hyperbilirubinemia reduces aging-associated inflammation and metabolic deterioration (e.g., glucose tolerance and mitochondrial dysfunction) [131]. These results suggest that plasma hemoglobin-mediated HO-1 induction and the consequential unconjugated hyperbilirubinemia has dual effects, depending on its concentration and duration of exposure. 

### 4.5. AD

AD is a neurodegenerative disease characterized by aggregated amyloid-β (Aβ) accumulation and tau pathology, leading to cognitive decline. Accumulating evidence suggests the involvement of the HO-1–CO axis in the regulation of AD pathogenesis. HO-1 upregulation and post-translational modifications (e.g., phosphorylation at serine residues and oxidative modification of HO-1) have been observed in the brains of patients with AD [132,133]. In mice, sustained HO-1 overexpression promotes tau aggregation in the brain by inducing tau phosphorylation [134]. In humans, iron-overload in the inferior temporal cortex may be involved in accelerated cognitive decline in patients with AD [135], suggesting that the HO-1-mediated iron accumulation may contribute to AD pathogenesis.

Iron-sequestering ferritin protein levels in the cerebrospinal fluid are reduced in prodromal disease progression in individuals with extensive Aβ pathology [136]. Ferroportin1 (a nonheme iron exporter) is reduced in the brains of transgenic mice (APPswe or PS1dE9) and patients with AD [137]. Neuronal deletion of ferroportin1 induces memory impairment by promoting ferroptosis and Aβ aggregation in AD mouse models [137]. In post-mortem AD brains and normal adult brain tissues, free Fe^3+^ induces aggregation of hyperphosphorylated tau. However, Fe^2+^ reverses this aggregation, consequently modulating the neurofibrillary tangle formation [138]. Nrf2 activation induces target genes, including *GPx4* and *SOD2* [139,140]. GSH is converted into GSSG by GPx4; CO, bilirubin, and ferritin are produced by HO-1; and superoxide is converted into hydrogen peroxide by SOD2 [140]. These antioxidant defense enzymes repair oxidative damage, thereby mitigating mitochondrial dysfunction and ferroptosis [100,101,102,103]. Since iron overload is correlated with AD pathogenesis [135], the suppression of ferroptosis by Nrf2-dependent induction of antioxidant enzymes (e.g., GPx4, SOD2, and HO-1) may provide a basis to explore their therapeutic potential in AD.

BVR-A is a pleiotropic enzyme involved in the reduction of biliverdin to the antioxidant bilirubin and in the secondary regulation of cell growth via its kinase activity. In the hippocampal samples from patients with AD and mild cognitive impairment, BVR-A expression and its oxidative or nitrosative modifications are increased, whereas its phosphorylation at serine, threonine, and tyrosine residues is decreased [141,142]. Given that the autophosphorylation of BVR-A increases its reductase activity, that is essential for the production of the antioxidant bilirubin [143], decreased levels of phosphorylated BVR-A are attributable to a reduction in cellular antioxidant activity in the cerebellum in AD and mild cognitive impairment. However, CORM-2 treatment or transient HO-1 induction protects cells from the Aβ-mediated oxidative damage in cultured astrocytes and neurons [144,145]. Hence, HO-1 metabolites, such as CO, bilirubin, and iron-mediated ferritin expression, may contribute to mitigating AD progress by suppressing oxidative stress and ferroptosis.

## 5. Conclusions

In brains injured by trauma, IR injury, or AD, long-term and high expression of HO-1 can facilitate cellular iron accumulation, leading to the induction of ROS’ and RNS’ production, inflammation, mitochondrial dysfunction, and ferroptosis. These processes are associated with neurodegeneration and neuronal dysfunction. However, lower and transient expression of HO-1 in the peri-injured brain regions promotes trapping of the redox-active free iron, avoiding the Fenton reaction and ROS generation via the induction of ferritin (an endogenous iron scavenger) and mitochondrial biogenesis. This exerts neurovascular protective effects, such as angiogenesis, neurogenesis, and mitochondrial biogenesis, by generating the cytoprotective and antioxidant metabolites, CO and bilirubin. In addition, a reciprocal positive feedback circuit between the HO–CO and NOS–NO pathways plays a key role in the amplification of intracellular signaling cascades involved in growth factor-mediated angiogenesis, mitochondrial biogenesis, and neurogenesis. In this review, we highlight the dual (i.e., beneficial and detrimental) roles of HO-1 and its metabolites in various neurovascular diseases, including AMD, IR injury, TBI, Gilbert’s syndrome, and AD (Table 1). This review also provides an up-to-date and comprehensive overview of the dual molecular mechanisms (i.e., neurovascular damage and regeneration) of HO-1 and its metabolites in neurovascular system pathology. Given the growing interest in the HO system as a potential therapeutic target, future studies should focus on the development of useful therapeutic strategies (i.e., the reduction of neurovascular inflammation and enhancement of neurovascular regeneration) and drugs that modulate the expression and activity of the HO system, predominantly HO-1. 

## Figures and Tables

**Figure 1 ijms-23-07041-f001:**
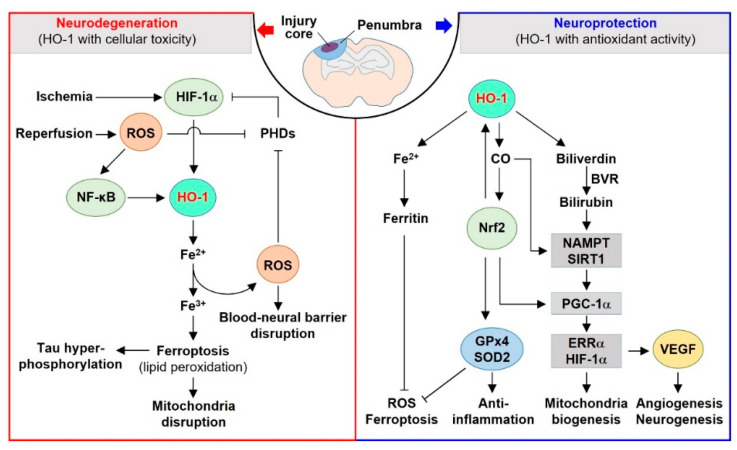
Dual effects of heme oxygenase (HO)-1-related signaling. In the injury core, long-term and high expression of HO-1 can lead to cellular iron accumulation, excessive reactive oxygen species (ROS)/reactive nitrogen species (RNS) production, proline hydroxylase (PHD) inactivation, hypoxia-inducible factor-1α (HIF-1α) stabilization, nuclear factor kappa-light-chain-enhancer of activated B cells (NF-κB) activation, inflammation, mitochondrial dysfunction, and ferroptosis, which are associated with neurodegeneration and blood–neural barrier disruption. However, lower and transient expression of HO-1 in the peri-injured regions of the brain promotes trapping of redox-active free iron, avoiding the Fenton reaction and ROS generation through induction of ferritin (an endogenous iron scavenger) and mitochondrial biogenesis, as well as exerting neurovascular protective effects, such as angiogenesis, neurogenesis, and mitochondrial biogenesis. In this neuroprotective phase, the interplay between HO-1 and nuclear factor erythroid 2-like 2 (Nrf2) may facilitate the anti-inflammatory actions of glutathione peroxidase 4 (GPx4) and superoxide dismutase 2 (SOD2). Further, carbon monoxide (CO) and bilirubin (converted from biliverdin by biliverdin reductase (BVR) may enhance mitochondrial biogenesis through the nicotinamide phosphoribosyl transferase (NAMPT)–sirtuin 1 (SIRT1)–peroxisome proliferator-activated receptor-γ coactivator-1α (PGC-1α)-mediated estrogen-related receptor α (ERRα)–HIF-1α axis and vascular endothelial growth factor (VEGF)-mediated angiogenesis and neurogenesis.

**Figure 2 ijms-23-07041-f002:**
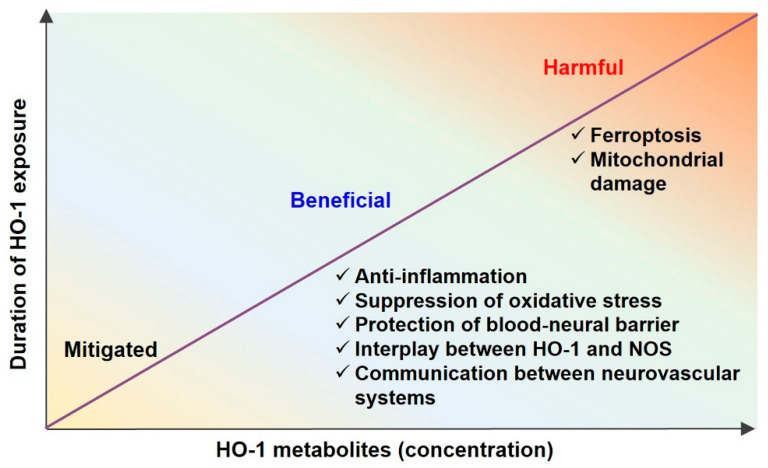
Effects of heme oxygenase (HO) metabolites according to concentration and duration of exposure. At low concentrations and short exposure durations, HO metabolites may be non-functional. At moderate concentrations and moderate exposure durations, HO metabolites may be beneficial, exhibiting anti-inflammatory effects, reduced oxidative stress, blood–neural barrier protection, stimulation of the interplay between HO-1 and nitric oxide synthase (NOS), and consequent communication between neurovascular systems. However, at high concentrations and long exposure durations, HO metabolites may be harmful by inducing ferroptosis and mitochondrial damage.

**Table 1 ijms-23-07041-t001:** Dual in vivo effects of HO-1 and its metabolites in various neurovascular diseases.

HO-1 and Its Metabolites (Function)	Summary (In Vivo)	Species	Disease	Ref.
HO-1 (beneficial)	HO- siRNA-treated retina demonstrates macrophage infiltration and severe destruction of the retinal structure.	Rat	IR injury	[112]
HO-1 (beneficial)	Gene transfer of HO-1 in IR activates BDNF-TrkB signaling pathway.	Rat	IR injury	[118]
HO-1 (detrimental)	Sustained HO-1 overexpression in transgenic mice facilitates tau aggregation in brains.	Mouse	AD	[134]
CO (beneficial)	CORM (ALF-186) is intravitreally applied into the left eyes of rats directly after retinal IR injury, resulting in enhanced retinal ganglion cells, reduction of inflammatory and apoptotic gene expression.	Rat	IR injury	[115]
CO (beneficial)	250 ppm CO is applied to MCAO mouse model, resulting in translocation of Nrf2 to nucleus, elevates HO-1 expression and reduced infarct size.	Mouse	IR injury	[51]
CO (beneficial)	CORM-3 reduces BBB leakage, pericyte cell death and oxidative stress-mediated HIF-1α expression and induces neurogenesis through activation of the NOS/HO pathway.	Mouse	TBI	[23]
CO (beneficial)	250 ppm CO treatment can prevent the pericyte cell death and promote neurogenesis.	Mouse	TBI	[23]
Bilirubin (beneficial)	Plasma bilirubin levels were increased on days 2, 3 and 4 in TBI patients, leading to an increase in antioxidant activity.	Human	TBI	[127]
Bilirubin (beneficial)	Soluble bilirubin nanoparticles protect mice from IR injury through attenuation of oxidative stress, apoptosis, and inflammation.	Mouse	IR injury	[113]
Bilirubin (detrimental)	Gilbert’s syndrome subjects demonstrate higher concentration of unconjugated bilirubin, carboxy hemoglobin and iron compared with control subjects.	Human	Gilbert’s syndrome	[132]
Iron (detrimental)	Increased cardioembolic stroke risk is related to increased serum iron and lower transferrin levels.	Human	IR injury	[110]
[96,97] Iron (detrimental)	More iron and transferrin in retinal pigment epithelium in AMD patients than age-mated controls.	Human	AMD	[98,99]
Iron (detrimental)	Iron deposition is increased during 3, 7, 14, 28 days in ipsilateral core-region.	Mouse	TBI	[121,122]
Iron (detrimental)	Iron overladed brains in the inferior temporal cortex may be involved in accelerated cognitive decline in AD patients.	Human	AD	[135]

*Abbreviations:* CORM, CO-releasing molecule; BDNF, brain-derived neurotrophic factor; TrkB, tropomyosin-related kinase B; MCAO, middle cerebral artery occlusion; AMD, age-related macular degeneration; IR, ischemia-reperfusion; TBI, traumatic brain injury; AD, Alzheimer’s disease.

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
