# Peer review of "Beneficial and Detrimental Roles of Heme Oxygenase-1 in the Neurovascular System"

_ijms, 2022, doi:10.3390/ijms23137041_

Round 1

Reviewer 1 Report

In this review article, the authors attempted to summarize the dual roles of heme oxygenase-1 (HO-1) and its relevance for several neurovascular diseases such as ischemia, brain injury, trauma, and aging. The authors provided a general overview about the heme oxygenase systems and the pathophysiological functions of their individual metabolites. The authors discussed the signaling cascades associated with harmful and beneficial effects. Overall, the manuscript is well written. However, it requires clarification and proofreading for typos and grammar. Below are some minor comments:

  1. The authors need to highlight the importance of this review over previously published work. The authors should write a paragraph about critical knowledge gaps, future perspectives and how the readers would benefit from this review.
  2. In the abstract, the authors mentioned that the review is about the dual roles of HO-1 and its metabolites, however, the title of Table 1 is about the in vivo effects of HO-1 or HO metabolites. Please clarify.
  3. The authors provided a summary of HO in the Introduction and then talked about the individual metabolites of HO but later abruptly switched to signaling cascade and dual roles of HO-1. The authors could improve the presentation style to make it easier for the readers to follow.
  4. The manuscript should be proofread for typos and grammar.

Author Response

Comment #1. The authors need to highlight the importance of this review over previously published work. The authors should write a paragraph about critical knowledge gaps, future perspectives and how the readers would benefit from this review.

Response to comment #1: We added the following sentences into the Conclusion section (Line 567-576). “In this review, we highlight the dual (i.e., beneficial and detrimental) roles of HO-1 and its metabolites in various neurovascular diseases, including AMD, IR injury, TBI, Gilbert’s syndrome, and AD (Table 1). This review also provides an up-to-date and comprehensive overview of the dual molecular mechanisms (i.e., neurovascular damage and regeneration) of HO-1 and its metabolites in neurovascular system pathology. Given the growing interest in the HO system as a potential therapeutic target, future studies should focus on the development of useful therapeutic strategies (i.e., reduction of neurovascular inflammation and enhancement of neurovascular regeneration) and drugs that modulate the expression and activity of the HO system, predominantly HO-1.”

Comment #2. In the abstract, the authors mentioned that the review is about the dual roles of HO-1 and its metabolites, however, the title of Table 1 is about the in vivo effects of HO-1 or HO metabolites. Please clarify.

Response to comment #2: We changed the title of Table 1 (Line 581) as follows. “Table 1. Dual in vivo effects of HO-1 and its metabolites in various neurovascular diseases.”  In addition, we have made changes through the manuscript to clarify.

Comment #3. The authors provided a summary of HO in the Introduction and then talked about the individual metabolites of HO but later abruptly switched to signaling cascade and dual roles of HO-1. The authors could improve the presentation style to make it easier for the readers to follow.

Response to comment #3: Based on your appreciated suggestion, we have made several changes to the manuscript to make it easier for the readers to follow.

Comment #4. The manuscript should be proofread for typos and grammar.

Response to comment #4: The revised manuscript has been corrected for typos and grammar with the help of a professional editor.

Reviewer 2 Report

The Authors tried to summarize recent evidence about HO-1 in the balance of vascular and energy-related pathophysiological processes. HO-1 could act as a pro- or anti-inflammatory protein and either stimulate the mitochondrial dynamics or impair them. The first figure and the table are very useful for delineating the hypothesized pathways and interactions and to read the different outcomes of significant studies all at a glance.

Briefly, in my opinion, the paper deserves to be published but some amendments are needed.

Figure 2. can be misleading, since it suggests a proper curve of exposure/concentration with a linear correlation and secondary plateau that has not been demonstrated. I suggest changing the graphical appearance.

The section on IR accounts for different organs and mechanisms and can be confusing, try to organize it to be more linear and readable. This suggestion can be considered also for the rest of the manuscript but the IR section needs it particularly.

The Authors stated: "Therefore, abnormal plasma or serum bilirubin levels may serve as a biomarker for the risk of cardiovascular disease and AD" however Gilbert syndrome is not associated with increased AD dementia. Moreover, the Authors did not mention this very specific disease for abnormal and chronic bilirubin levels that is common in humans and could be used to prove or disprove some of the proposed mechanisms with a reliable pathological model. I suggest discussing Gilbert syndrome and the implication of bilirubin levels in a separate paragraph.

Author Response

Comment #1: Figure 2. can be misleading, since it suggests a proper curve of exposure/concentration with a linear correlation and secondary plateau that has not been demonstrated. I suggest changing the graphical appearance.

Response to comment #1: We changed it with a linear correlation (Line 93) per reviewer’s suggestion.

Comment #2: The section on IR accounts for different organs and mechanisms and can be confusing, try to organize it to be more linear and readable. This suggestion can be considered also for the rest of the manuscript but the IR section needs it particularly.

Response to comment #2: Based on reviewer’s suggestion, the IR section has been re-organized and changed as follows (Line 415-431). The changes are indicated in red ink. “Beneficial effects of HO-1 and its metabolites in IR injuries have been reported. HO-1 is upregulated in rat Müller cells of the retina and promotes cell survival after IR injury, which is abolished by the intravitreal injection of HO-1 siRNA before ischemia [115]. HO-1 siRNA injection increases macrophage infiltration and severe destruction of the retinal structure in a rat model of IR injury [115]. Moreover, HO-1 gene delivery exhibits the most potent suppressive effects on the expression of TLR-4, TLR-9, and inflammatory cytokines in post-ischemic limb tissue [116], highlighting the anti-inflammatory role of HO-1 and its metabolites in ischemic tissue. Compared to WT mice, HO-1 KO mice exhibit greater infiltration of inflammatory cells into the ischemic gastrocnemius muscle [116]. This indicates that HO-1 deficiency predisposes an exaggeration of immune cell infiltration and inflammatory responses, potentially by upregulating cell adhesion molecules in the vasculature. Further, HO-1-mediated antioxidant properties, mediated by the generation of CO and bilirubin, may contribute to its beneficial effects on ischemic injury. Administration of soluble bilirubin nanoparticles using polyethylene glycol‐modified bilirubin exerts potential therapeutic effects in a mouse model of IR injury [117]. Mice with depleted endogenous bilirubin via BVR deletion are susceptible to excitotoxic neuronal death due to a decrease in antioxidant activity, particularly to superoxide radicals [118].”

Comment #3: The Authors stated: "Therefore, abnormal plasma or serum bilirubin levels may serve as a biomarker for the risk of cardiovascular disease and AD" however Gilbert syndrome is not associated with increased AD dementia. Moreover, the Authors did not mention this very specific disease for abnormal and chronic bilirubin levels that is common in humans and could be used to prove or disprove some of the proposed mechanisms with a reliable pathological model. I suggest discussing Gilbert syndrome and the implication of bilirubin levels in a separate paragraph.

Response to comment #3: We made changes (Line 143-145 & Table 1) and added a new sub-section for Gilbert’s syndrome (Line 505-516) per your suggestion. The changes are indicated in red ink.

First, on line 143-145: “In this regard, abnormal plasma or serum bilirubin levels may serve as a biomarker for the risk of cardiovascular disease, Gilbert’s syndrome, and AD [30-32].”

Second, on line 581: we added new information in red into Table 1.

Finally, om line 505-516: we added following new section. “4.4. Gilbert’s syndrome: Hyperbilirubinemia is classified as having predominantly unconjugated bilirubin (indirect bilirubin) or conjugated bilirubin (direct bilirubin). Neonatal hyperbilirubinemia results from increased total serum bilirubin and clinically presents as pathologic conditions, such as jaundice, as well as behavioral and neurological impairments [132]. Gilbert’s syndrome is characterized by elevated unconjugated bilirubin levels in the blood due to increased bilirubin production or decreased bilirubin clearance [32, 132]. Bilirubin-induced neurotoxicity may occur when the central nervous system is chronically exposed to high levels of unconjugated bilirubin. Indeed, patients with Gilbert’s syndrome have higher concentrations of unconjugated bilirubin, carboxyhemoglobin and iron than healthy controls [133]. This suggests that Gilbert’s syndrome may be underpinned by plasma hemoglobin-mediated HO-1 induction and consequential unconjugated hyperbilirubinemia.”

Round 2

Reviewer 2 Report

The Authors modified the IR section marginally without avoiding the mixture of different organs from the retina to a skeletal muscle to the retina again and the CNS/heart, which can be confusing. Moreover, the introduction of Gilbert is substantially formal, my comment considered that abnormal levels of bilirubin exist in these patients, however, they have no increased risk of cardiovascular disease and/or AD as the authors stated. The authors should provide references and add comments about how these clinical data could not be in conflict with their proposed mechanism. Finally, they wrote: "This suggests that Gilbert’s syndrome may be underpinned by plasma hemoglobin-mediated HO-1 induction and consequential unconjugated hyperbilirubinemia" but the defect is the uridine diphosphate glucuronosyltransferase 1A1 enzyme.

Author Response

Comment #1: The Authors modified the IR section marginally without avoiding the mixture of different organs from the retina to a skeletal muscle to the retina again and the CNS/heart, which can be confusing.

Response to comment #1: To make consistency related to organs associated with central neurovascular system, we removed several sentences related to peripheral neurovascular system. Thank you for your valuable comment.

Comment #2: Moreover, the introduction of Gilbert is substantially formal, my comment considered that abnormal levels of bilirubin exist in these patients, however, they have no increased risk of cardiovascular disease and/or AD as the authors stated. The authors should provide references and add comments about how these clinical data could not be in conflict with their proposed mechanism.

Response to comment #2: As you advised, we toned down our proposed mechanism about abnormal levels of bilirubin regarding to cardiovascular disease and/or AD. It would be better to remove “cardiovascular disease and/or AD” as follows.

(Line 143-144) “Thus, abnormal plasma or serum bilirubin levels may serve as a biomarker for the risk of developing Gilbert’s syndrome [30].”

Comment #3: Finally, they wrote: "This suggests that Gilbert’s syndrome may be underpinned by plasma hemoglobin-mediated HO-1 induction and consequential unconjugated hyperbilirubinemia" but the defect is the uridine diphosphate glucuronosyltransferase 1A1 enzyme.

Response to comment #3: We appreciate your valuable comment. Therefore, we modified the paragraphs and added new information as follows.

(Line 494-514)

“4.4. Gilbert’s syndrome

Hyperbilirubinemia is classified as having predominantly unconjugated bilirubin (indirect bilirubin) or conjugated bilirubin (direct bilirubin). Neonatal hyperbilirubinemia results from increased total serum bilirubin and clinically presents as pathologic conditions, such as jaundice, as well as behavioral and neurological impairments [129]. Gilbert’s syndrome is characterized by elevated unconjugated bilirubin levels in the blood due to increased bilirubin production or decreased bilirubin clearance [30, 129]. The unconjugated bilirubin is transformed to conjugated bilirubin by the activity of hepatic uridine diphosphate-glucuronosyltransferase 1A1 (UGT1A1) to increase its water solubility [130]. Defective UGT1A1 can be a cause of Gilbert’s syndrome.

Bilirubin-induced neurotoxicity may occur when the central nervous system is chronically exposed to high levels of unconjugated bilirubin. Indeed, patients with Gilbert’s syndrome have higher concentrations of unconjugated bilirubin, carboxyhemoglobin and iron than healthy controls [131]. Nonetheless, HO-1-mediated bilirubin production may have a beneficial role in various pathologic conditions [130]. Moderate hyperbilirubinemia reduces aging-associated inflammation and metabolic deterioration (e.g., glucose tolerance and mitochondrial dysfunction) [132]. These results suggest that plasma hemoglobin-mediated HO-1 induction and consequential unconjugated hyperbilirubinemia has dual effects depending on its concentration and duration of exposure.”
